# Systematic review and integrated data analysis reveal diverse pangolin-associated microbes with infection potential

Run-Ze Ye [1,5], Xiao-Yang Wang[1,5], Yu-Yu Li[1,2,5], Bao-Yu Wang[1,5], Ke Song[1], Yi-Fei Wang[1], Jing Liu[1], Bai-Hui Wang[1], Shan-Shan Wang[1], Qing Xu[1], Ze-Hui Li[1], Yi-Di Du[1,2], Jin-Yue Liu[1,2], Jia-Jing Zheng[2], Li-Feng Du[1], Wenqiang Shi[2], Na Jia [2,3], Jia-Fu Jiang [2,3], Xiao-Ming Cui[2,3] ✉, Lin Zhao[1,4] ✉ & Wu-Chun Cao [1,2,3] ✉

There has been increasing global concern about the spillover transmission of pangolin-associated microbes. To assess the risk of these microbes for emergence as human pathogens, we integrated data from multiple sources to describe the distribution and spectrum of microbes harbored by pangolins. Wild and trafficked pangolins have been mainly recorded in Asia and Africa, while captive pangolins have been reported in European and North American countries. A total of 128 microbes, including 92 viruses, 25 bacteria, eight protists, and three uncharacterized microbes, have been identified in five pangolin species. Out of 128 pangolin-associated microbes, 31 (including 13 viruses, 15 bacteria, and three protists) have been reported in humans, and 54 are animal-associated viruses. The phylogenetic analysis of human-associated viruses carried by pangolins reveals that they are genetically close to those naturally circulating among human populations in the world. Pangolins harbor diverse microbes, many of which have been previously reported in humans and animals. Abundant viruses initially detected in pangolins might exhibit risks for spillover transmission.

Pangolins (Class Mammalia, Order Pholidota) are the most-poached and illegally trafficked mammals in the world due to the significant consumer demand for their scales and meat[1]. According to new data from TRAFFIC, around one million pangolins have been poached in the last decade, and more than 23.5 tons of pangolins or their products have been trafficked in 2021 alone[2]. Natural habitat deterioration and climate change have also dramatically shrunk the distribution of pangolins worldwide[3]. All eight pangolin species are heading to the edge of extinction[4]. With the exhaustion of wild populations, captive breeding is becoming an important and well-accepted way of saving pangolins[5]. Various human and animal

pathogens have recently been detected in wild, smuggled, and captive pangolins[1,6–9], suggesting them as reservoir hosts of emerging infectious diseases. The recent discovery of SARS-CoV-2-related coronaviruses in Malayan pangolins (*Manis javanica*) has provoked global public health concern[10]. Pangolins are found to be susceptible to infections because of their compromised innate immunity[1]. Furthermore, the illegal trade increases the risk of interspecies transmission of pathogens between pangolins and other mammals[6]. Therefore, it is important to discern the pangolin-associated microbes and pathogens for risk assessment of cross-species transmission and future epidemics. In this study, we integrated data from

[1]Institute of EcoHealth, School of Public Health, Cheeloo College of Medicine, Shandong University, Jinan, P. R. China. [2]State Key Laboratory of Pathogen and Biosecurity, Beijing Institute of Microbiology and Epidemiology, Beijing, P. R. China. [3]Research Unit of Discovery and Tracing of Natural Focus Diseases, Chinese Academy of Medical Sciences, Beijing, P. R. China. [4]Department of Epidemiology, School of Public Health, Cheeloo College of Medicine, Shandong University, Jinan, P. R. China. [5]These authors contributed equally: Run-Ze Ye, Xiao-Yang Wang, Yu-Yu Li, Bao-Yu Wang. ✉e-mail: cuixm7@163.com; zhaolin1989@sdu.edu.cn; caowuchun@126.com

multiple sources to characterize the distribution and spectrum of microbes harbored by pangolins, to understand potential pathogens infectious to humans and animals, and to provide evidence for assessing the risk of these pangolin-associated microbes for emergence in human populations.

## Results

### Characteristics of pangolin database

After removing duplication, the analysis of integrated data collected from multiple sources, including literature review and related websites, revealed that a total of 2337 records with locations of different pangolin species were reported in 60 countries around the world, including 734 Chinese pangolin (*Manis pentadactyla*) records, 617 Indian pangolin (*Manis crassicaudata*) records, 479 Malayan pangolin (*Manis javanica*) records, 13 Palawan pangolin (*Manis culionensis*) records, 78 long-tailed pangolin (*Phataginus tetradactyla*) records, 215 tree pangolin (*Phataginus tricuspis*) records, 91 giant pangolin (*Smutsia gigantea*) records, and 110 ground pangolin (*Smutsia temminckii*) records (Fig. 1a). Among them, there were 1854 records of wild pangolins, 279 records of trafficked pangolins, 197 records of captive pangolins, and seven records for the unknown origin of pangolins. Detailed results of the integrated data are described in the appendix (Supplementary Text 1). Of the 142 microbial records extracted from the publication, 128 with clear geographic location were included in the 2337 records for distribution analysis, while the other 14 were not included due to unavailable information on their geographic location. Among 142 microbial records, 112 were associated with viruses, 11 with bacteria, 13 with protists, and six were records with zero positive rates. A total of 461 sequences, including 354 virus sequences, 89 bacterium sequences, and 18 protist sequences, were obtained from databases of GenBank, Global Initiative on Sharing All Influenza Data (GISAID), and National Genomics Data Center (NGDC) after removing duplication.

Although a paper about pangolins was published as early as 1888, the study on the specific animal has generally been neglected until the 21st century. Notably, pangolin-associated microbes had never been reported before 1976 (Fig. 1b). After SARS-CoV-2-related coronaviruses were reported in *M. javanica* in 2020[10], the publications regarding either pangolins or their associated microbes sharply increased. The sequences of microbes harbored by pangolins were released mostly in the recent few years.

### Geographical distribution of pangolins

The natural habitats of pangolins were located between the northern and southern hemispheres, ranging from 40° N to 30° S latitudes (Fig. 2a). Wild pangolins were predominantly distributed in 47 coastal countries of Asia and Africa (Fig. 2a). The trafficked pangolins were reported in 23 countries, all of which were also the original countries of wild pangolins. Notably, captive pangolins in zoos, rescue centers, or breeding farms were widely reported around the world, especially in European and North American countries, where no wild pangolins exist (Fig. 2b). *M. pentadactyla* was distributed in Asian countries, mostly in China and Nepal, and often was illegally smuggled into India (Supplementary Fig. 1). *M. javanica* was most prevalent in Southeast Asian countries, such as Malaysia, Vietnam, Thailand, and Indonesia, but smuggling mostly occurred in Myanmar (Supplementary Fig. 1). *M. crassicaudata* was present in India and its adjacent countries, and most abundant in Sri Lanka (Supplementary Fig. 1), where pangolins were widespread in the wild and had been heavily trafficked. *M. culionensis* was only distributed in the Philippines (Supplementary Fig. 2). *P. tricuspis* and *S. gigantea* were mainly distributed in the western coastal areas and middle part of Africa, while *S. temminckii* was more concentrated in the eastern and southern African countries (Supplementary Figs. 3–5). *P. tetradactyla* was only found in Ghana (Supplementary Fig. 6).

### Diversity and distribution of pangolin-associated microbes

A total of 128 microbes belonging to 46 families and 33 orders were proved in pangolins (Fig. 3a), including 92 virus species, 25 bacterium species, eight protist species, and three pangolin-associated microbes (two bacteria and a protist) that were not classified to species in the original reports. The microbes were detected from different kinds of pangolin samples, 42 of which were from mixed tissues, and various samples, including blood, sera, lungs, livers, spleens, hearts, muscles, feces, anal swabs, and throat swabs (Supplementary Table 1). Among the 128 microbes, 70 (including 64 viruses, three bacteria, and three protists) were identified in both literature and sequence databases, 37 (including 27 viruses, six bacteria, and four protists) were reported only in the literature, and 21 (including one virus, 18 bacteria, and two protists) appeared only in the sequence databases (Supplementary Table 2). Among the 461 microbial sequences (including 354 viral, 89 bacterial, and 18 protist sequences), 112 sequences being associated with 19 families (three viruses, 14 bacteria, two protists) were submitted directly to the above three databases without publication, which enriched the description of the microbe spectrum of pangolins. The GenBank, GISAID, and NGDC accession numbers and detailed information on the sequences of pangolin-associated microbes are shown in the appendix (Supplementary Table 3).

Pangolin-associated microbes were mainly reported in countries of South Asia, Southeast Asia, and the middle part of Africa (Fig. 3b). Viruses and bacteria carried by pangolins were only detected in Asian countries, such as Vietnam and Thailand and were most abundant in southern China. Protists were mainly found in Asian and African countries, such as southeastern China, India, Bangladesh, Malaysia, and African countries. Microbes carried by wild pangolins were mainly reported in Asian countries, including eastern and southwestern China and Malaysia. Microbes in trafficked pangolins were only detected in Asian countries, such as Vietnam, Thailand, and Malaysia, and were most abundant in southern China. Microbes carried by captive pangolins were found in Asia, such as southeastern China, India, Bangladesh, and Malaysia. Although the presence of wild or captive pangolins has been documented in many countries, there has been no report on microbe infections among pangolins in these countries, such as Sri Lanka, Pakistan, as well as some countries in South Africa and Europe.

### Positive rate and pathogenicity of pangolin-associated microbes

We analyzed the pangolin-associated microbes in relation to species of pangolins. *M. javanica* carried the most diverse microbes, followed by *M. pentadactyla*, *M. crassicaudata*, *P. tricuspis* and *P. tetradactyla*. In addition, six sequences of five pangolin-associated microbes were obtained from GenBank, but the information regarding pangolin species was unavailable (Fig. 4). Viruses were only identified in *M. javanica*, *M. pentadactyla* and some unknown species of pangolins. Bacteria were detected mostly in *M. javanica*, *M. crassicaudata* and a few unknown species of pangolins. Protists were found in *M. javanica*, *P. tetradactyla*, *P. tricuspis* and unknown species of pangolins. No microbe was reported in *M. culionensis*, *S. temminckii* and *S. gigantea*.

We compared the distribution of microbes harbored by wild, trafficked, and captive pangolins (Fig. 4). A total of 22 microbes, including 12 viruses, five bacteria, and five protists, were associated with four species of wild pangolins. Trafficked pangolins, including *M. javanica* and *M. pentadactyla*, carried at least 84 microbes, involving 80 species of viruses, three species of bacteria, and an uncharacterized *Babesia*. Only three microbes, including a virus, a bacterium, and one species of protist, were identified in two species of captive pangolins. To investigate the public health and veterinary importance of pangolin-associated microbes, we categorized them into four types (Fig. 4). Among the 128 pangolin-associated microbes, 31 (including 13 viruses, 15 bacteria, and three protists) have been reported in humans. Fifty-nine (including 53 viruses, three bacteria, and three protists) were initially discovered from pangolins. Fifty-four (including 35 viruses, 14

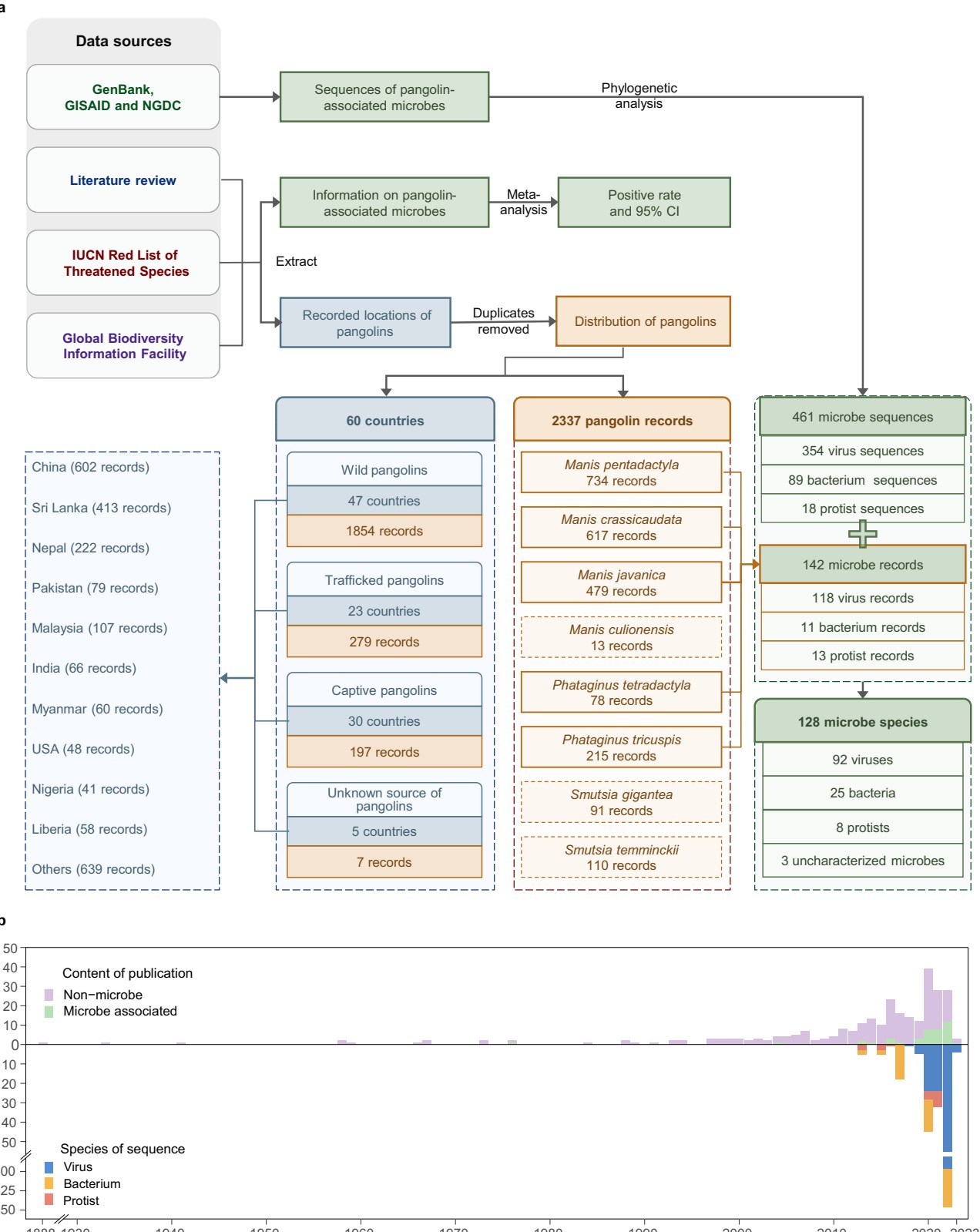

**Fig. 1 | Study design and data sources. a** The database of pangolins was collected and integrated from multiple data sources, including literature review, related websites, GenBank, Global Initiative on Sharing All Influenza Data (GISAID), and National Genomics Data Center (NGDC), and a comprehensive analysis was conducted. **b** Number of the publications and sequences available in each year. At the top of the x-axis is the number of publications about pangolins published each year, and at the bottom of the x-axis is microbe sequences by year of microbial sequence databases release. Source data are provided as a Source Data file.

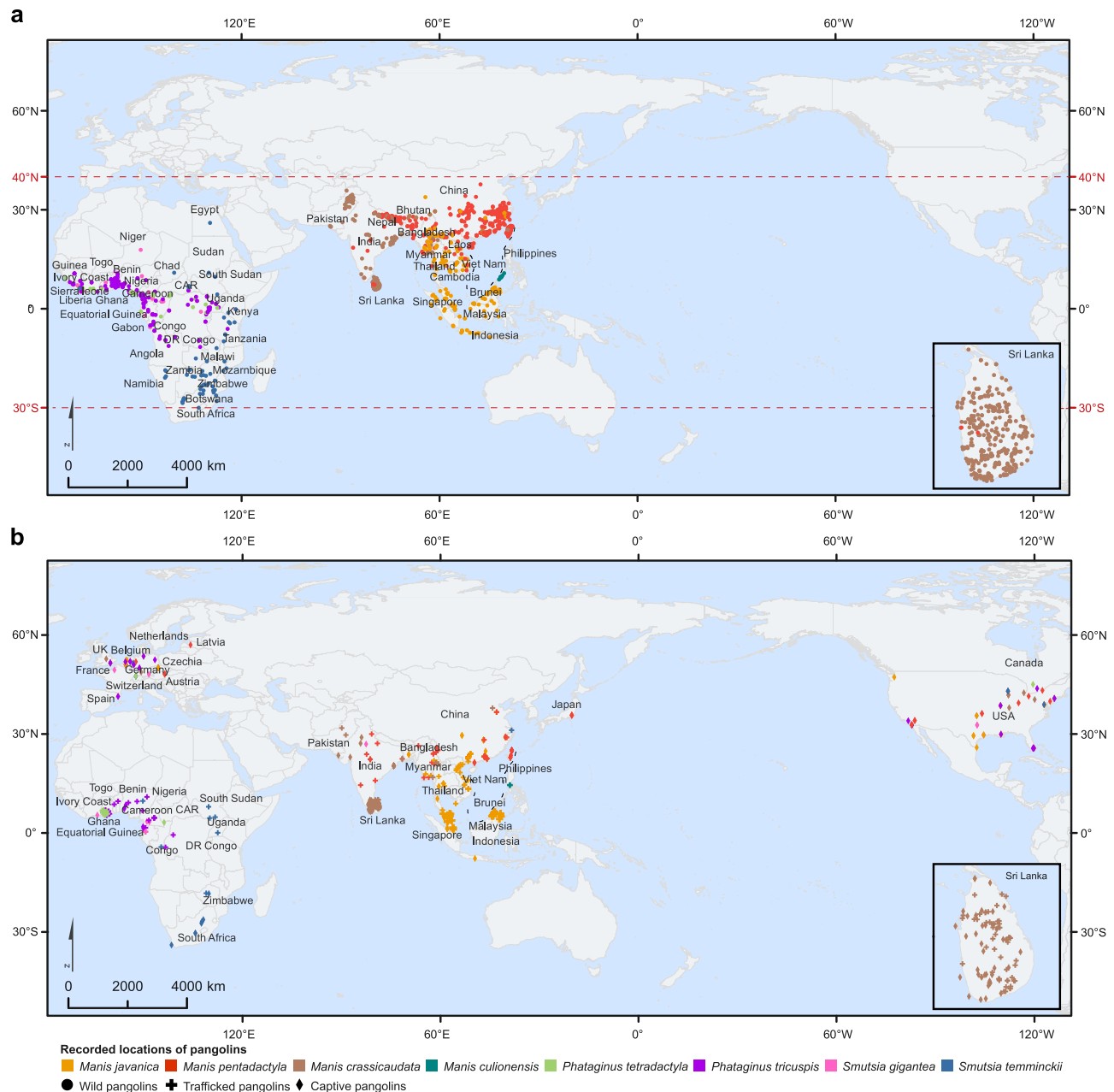

**Fig. 2 | Geographical distribution of pangolins. a** Recorded locations of wild pangolins. Wild pangolins were distributed in areas between 40° N and 30° S latitudes. **b** Recorded locations of trafficked and captive pangolins. Source data are provided as a Source Data file.

bacteria, and five protists) had been previously reported in other animals. Among the 54 microbes found in other animals, 25 were zoonotic and have been included in the above 31 human pathogens. There were five microbes had been deposited in GenBank with unknown hosts except for pangolins. Notably, ten of 13 human-associated viruses (76.92%) were identified in trafficked pangolins. Two mosquito-borne human pathogens, *Japanese encephalitis virus* (JEV) and *Chikungunya virus*, were only reported in captive pangolins.

In the meta-analysis (Fig. 4), the family *Papillomaviridae* had the highest positive rate among all viruses (48.33%; 95% confidence interval [CI]: 39.49–59.15%) followed by families *Pneumoviridae* (37.54%; 95% CI: 0.19–88.69%) and *Parvoviridae* (32.70%; 95% CI: 16.28–65.66%) (Supplementary Fig. 7). Viruses carried by *M. javanica* mainly belonged to families *Pneumoviridae*, *Flaviviridae*, and *Coronaviridae*, such as *human orthopneumovirus* (HRSV, 50.03%; 95% CI 0.01–99.99%), *Dongyang pangolin virus* (DYPV, 70.73%; 95% CI

45.35–91.65%), *pangolin coronavirus* (pangolin-CoV, 12.49%; 95% CI 1.08–23.90%), and SARS-CoV-2-related coronavirus (16.50%; 95% CI 1.58–38.95%) (Supplementary Table 4). *M. javanica* also carried a variety of viruses belonging to the family *Parvoviridae*, such as *Carnivore protoparvovirus 1* (33.70%; 95% CI 4.56–69.69%). Most species of bacteria were only found in *M. javanica*, among them *Aeromonas dhakensis* (100%), *Ehrlichia ruminantium* (50%), and *Candidatus* Anaplasma *pangolinii* (53.33%) had high positive rates (Supplementary Fig. 8). *Morganella morganii* was detected in both *M. javanica* and *M. pentadactyla*, with a positive rate of 100% (Supplementary Fig. 8). A few protists, such as *Eimeria nkaka* (100%) and Uncharacterized *Babesia* (41.41%), were common in *P. tricuspis* and *M. javanica* (Supplementary Fig. 8). To explore the reason for heterogeneity in the infection rate of each microbe among publications, we performed the meta-regression analysis and did not find any factor associated with the heterogeneity (Supplementary Tables 5 and 6).

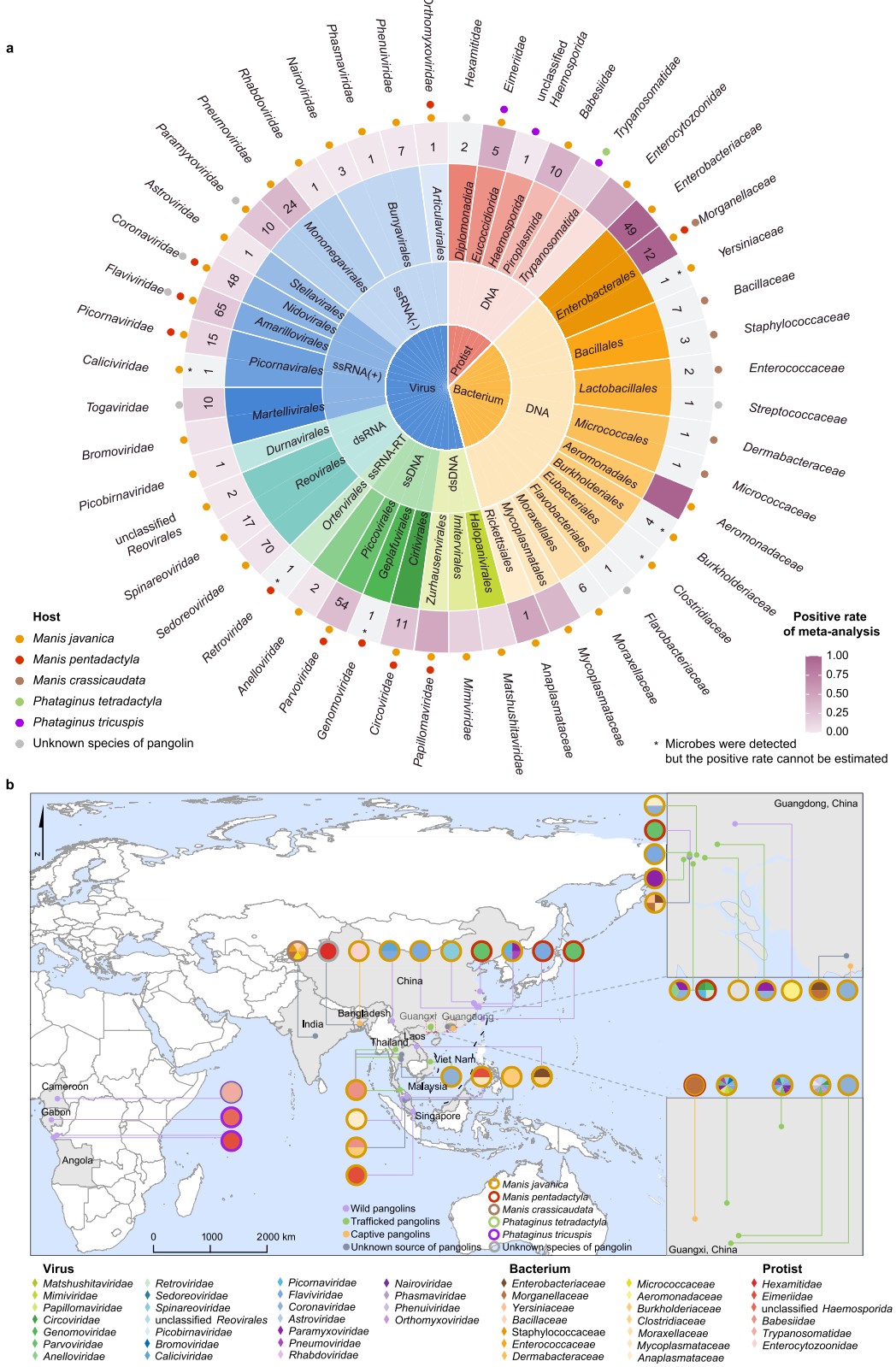

**Fig. 3 | Classification and geographical distribution of microbes carried by different pangolins. a** Classification of microbes carried by pangolins. The inner circle divides all microbes into three categories: viruses, bacteria, and protists. The second circle shows the molecular types of microbes. The third circle represents the order to which the microbe belongs. The outermost names represent families of microbes. In the grid of the fourth circle, the number of sequences retrieved from each family is marked with numbers, and the positive rates of microbes carried by pangolins are represented by a gradient color. The point outside the fourth circle displays the host, which is colored according to pangolin species:

orange, *Manis javanica*; red, *M pentadactyla*; brown, *M crassicaudata*; green, *Phataginus tetradactyla*; violet, *P tricuspis*; gray, unknown species of pangolin. **b** Geographical distribution of pangolin-associated microbes. Countries with pangolin-associated microbe records are annotated in light gray. The pies indicate microbe composition, with the color of circle outlines representing pangolin species: orange, *M. javanica*; red, *M pentadactyla*; brown, *M crassicaudata*; green, *P. tetradactyla*; violet, *P tricuspis*; gray, unknown species of pangolin. Source data are provided as a Source Data file.

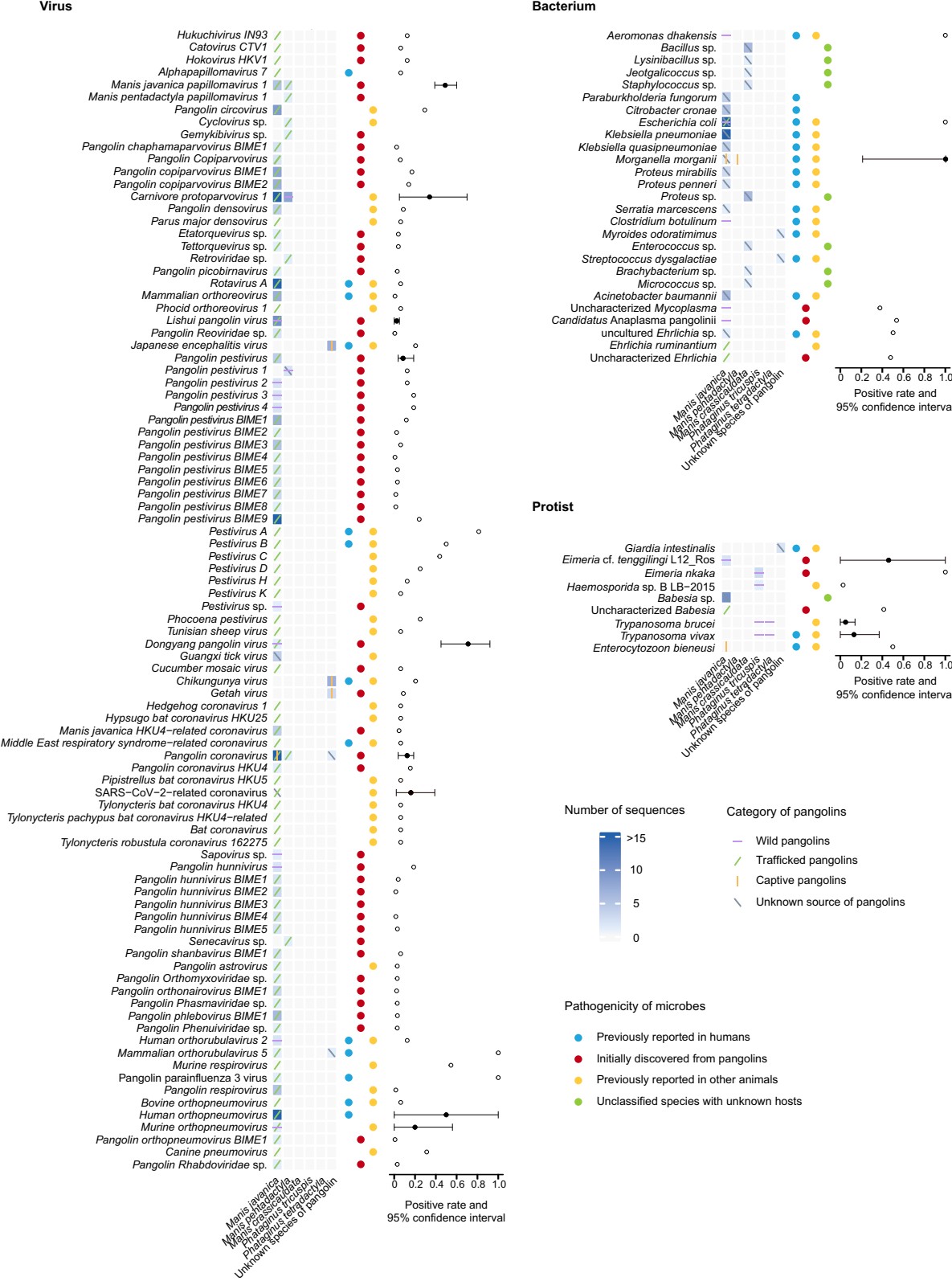

**Fig. 4 | Prevalence and pathogenicity of microbes in pangolins with different characteristics.** The columns from left to right in the heat map are pangolin species, and the colors of each grid in the heat map represent the number of sequences. The colored lines in each grid represent the categories of pangolin from which the microbe was detected. Violet horizontal lines represent wild pangolins, green oblique lines represent trafficked pangolins, orange vertical lines represent captive pangolins, and dark gray oblique lines represent unknown source of pangolins. The pathogenicity of each microbe is listed on the right side of the heat map, including humans (blue), pangolins (red), other animals (yellow), and unknown (green) pathogenic. Positive rate and 95% confidence interval (CI) for each microbe are provided at the end of each row. The error bars represent the 95% CIs of the positive rates. The center points for the error bars represent the positive rates calculated by the meta-analysis. The open circles represent a single positive rate without a 95% CI. The study size and positive rate with 95% CI are provided in Supplementary Table 4. Source data are provided as a Source Data file.

## Phylogeny of human-associated viruses in pangolins

We did phylogenetic analyses of eight pangolin-associated human viruses, whose sequences were available in databases, to understand their evolutionary characteristics. Among the eight human viruses, five were responsible for respiratory infections, one for gastrointestinal infection, and two for mosquito-borne infections. Human respiratory syncytial virus (HRSV), which belongs to the genus *Orthopneumovirus*, is the leading cause of severe respiratory disease in children under two years of age[11]. Our phylogenetic analyses based on RNA-dependent RNA polymerase (RdRp) proteins revealed that HRSV-A and HRSV-B in pangolins were in the same clades as those currently circulating in humans, with aa identities of 99.69–99.95% and 99.86–99.95%, respectively. In addition, a new species of the genus *Orthopneumovirus* in pangolins was genetically related to human HRSV-A[6] (Fig. 5a). *Mammalian orthoreovirus* which is widely distributed in mammals including humans, was also detected in a *M. javanica* with closely related to that detected in human from Switzerland (Fig. 5b). Pangolin parainfluenza 3 virus (Fig. 5c) and *human orthorubulavirus 2* (Fig. 5d) identified in *M. javanica* were in the same clusters with those present in humans worldwide[7, 12]. *Rotavirus A* in the genus *Rotavirus* that can cause diarrhea in children and animals[13], was clustered in a separate branch in the phylogenetic tree (Fig. 5e). JEV, an arbovirus belonging to the family *Flaviviridae*, could accidentally infect humans[14]. JEVs detected in captive pangolins were genetically clustered in the same clade as those from bats and mosquitos in the same country (Fig. 5f). *Chikungunya viruses* in pangolins fell into two distinct branches[14], one of which was clustered with those from southeastern Asia, and the other was genetically close to those from mosquitos in Senegal of western Africa (Fig. 5g). During the COVID-19 pandemic, a series of studies[9, 15] reported coronavirus of the genus *Betacoronavirus* in pangolins, belonging to either *Sarbecovirus* or *Merbecovirus* subgenus, respectively. Their phylogenetic relationship is displayed in the appendix (Supplementary Figs. 9–14).

## Discussion

Wild pangolins have been found to inhabit between 40° N and 30° S latitudes, particularly in coastal countries of Asia and Africa. Increasing demand of human for pangolin meat and scales has led to pangolins becoming the world's most trafficked wild mammal[16]. The local cultures, religions, economic development levels, living habits, laws, and policies have determined the extent of illegal pangolin trade and might explain why *M. javanica* and *M. pentadactyla* are most heavily trafficked in Asia and the illegal smuggling has increased in Africa[17]. Up to 65% of seized pangolins and related products have been smuggled by sea[18], and that might explain why trafficked pangolins are usually found in coastal countries (Fig. 2). The massive poaching and trafficking have greatly decreased the wild pangolin population. Artificial breeding has thus been suggested as a vital approach to protecting the endangered animals[5]. Because pangolins rely heavily on the natural ecosystem, there remain many obstacles to successfully breeding pangolins. It turns out that artificial pangolins often die of diseases and have obviously lower survival rates than wild pangolins held in captivity[19]. Therefore, a complete prohibition of the pangolin trade is crucial to pangolin conservation.

The integrated data revealed that pangolins harbor at least 128 microbes. Due to the limited detection technology, early surveys mainly focused on the protists carried by pangolins. Since SARS-CoV-2-related coronaviruses were discovered in *M. javanica* in 2020[9], up to 86 species of viruses were detected in pangolins in recent years, mostly through next-generation sequencing or RT-PCR tests of pangolin samples. The great number and diversity of pangolin-associated microbes indicate that pangolins might be vulnerable to infections with various microbes because the gene of interferon epsilon (*IFNE*), which can establish a first-line innate immune defense against pathogens in most placental mammals, is pseudogenized in pangolins[1].

These findings suggest that pangolins might be developed as suitable animal hosts and contribute to the emergence or re-emergence of various pathogens. Notably, about 86.96% (80 of 92) viruses were identified in smuggled pangolins, suggesting that the illegal trade with bad transportation conditions has facilitated cross-species transmission of the viruses among pangolins and other animals, including humans. The illegal pangolin trade not only threatens the survival of pangolin populations and ecosystem integrity but also brings several public and veterinary health consequences, such as the spreading of zoonotic pathogens into new geographical areas.

The findings of phylogenetical analyses of eight viruses reported in humans carried by pangolins indicate that there is no genetic distinction from those naturally circulating among human populations in the world, suggesting that these viruses might be transmitted from humans to pangolins, especially for the respiratory-transmitted viruses[8, 12]. Two mosquito-borne viruses, JEV and *Chikungunya virus*, are detected in captive pangolins. Although no direct evidence for vector transmission is available, phylogenetic analysis results indicate that they are clustered with the strains from mosquitos. A comparable example is DYPV, a new species in the genus *Pestivirus* detected in pangolins, which might cause hemorrhagic disease. The virus was also present in nymph ticks collected from the pangolins[12], indicating the risk for transmission of pangolin-associated viruses by tick bites.

This study has several limitations. First, only English and Chinese literature were included in our study, and pangolin records in other languages could have been missed. Second, the pangolin species without reported pathogens currently might still have microbes that have not been detected before, and variations in detection technology could have introduced bias into the positive rate of pathogens. Third, the controverse taxonomy of pangolins due to local names used in the early literature might impact the geographical distribution objectivity of some pangolin species. Finally, a lot of early studies have provided limited species information about the microbes detected, preventing us from accurately classifying them.

In conclusion, pangolins, especially the trafficked *M. javanica* and *M. pentadactyla*, harbor a variety of microbes, becoming a great threat to human and animal health. Except for Asian and African countries, where pangolins are mainly distributed in and often smuggled into, pangolins are also bred in captivity in Europe and North America and are suitable to survive in more extensive areas. Multidepartment collaboration and mass participation should be strengthened to protect pangolins, as well as human and planetary health. Authorities should enact laws and regulations to combat the illegal pangolin trade, and the punishments for such crimes should be severe. Enhanced surveillance and exploration are warranted to better understand the full pathogen spectrum of all pangolin species and to mitigate the threat of emerging zoonotic diseases spread by pangolins. From the One Health perspective, the health of pangolins is closely linked to the health of people and ecosystems. The global ban on pangolin trade should be strictly implemented, and governments should take measures to protect the natural habitat of pangolins. Veterinary workers and medical personnel are urged to be vigilant against zoonotic pathogens and diseases spread by pangolins.

## Methods

### Data collection

Data were collected from multiple sources, including literature review, related websites, and GenBank (Fig. 1a). Data from publications were extracted by two independent reviewers (X.Y.W. and B.Y.W.) through searching electronic databases (PubMed, China National Knowledge Infrastructure, and the WanFang database) for studies published before February 16, 2023. We searched the above databases separately with the terms of Latinized (Linnaean) binomial names and alternative names of each pangolin species (in English or Chinese). To avoid controverse taxonomy of pangolins, we utilized the specific Latin

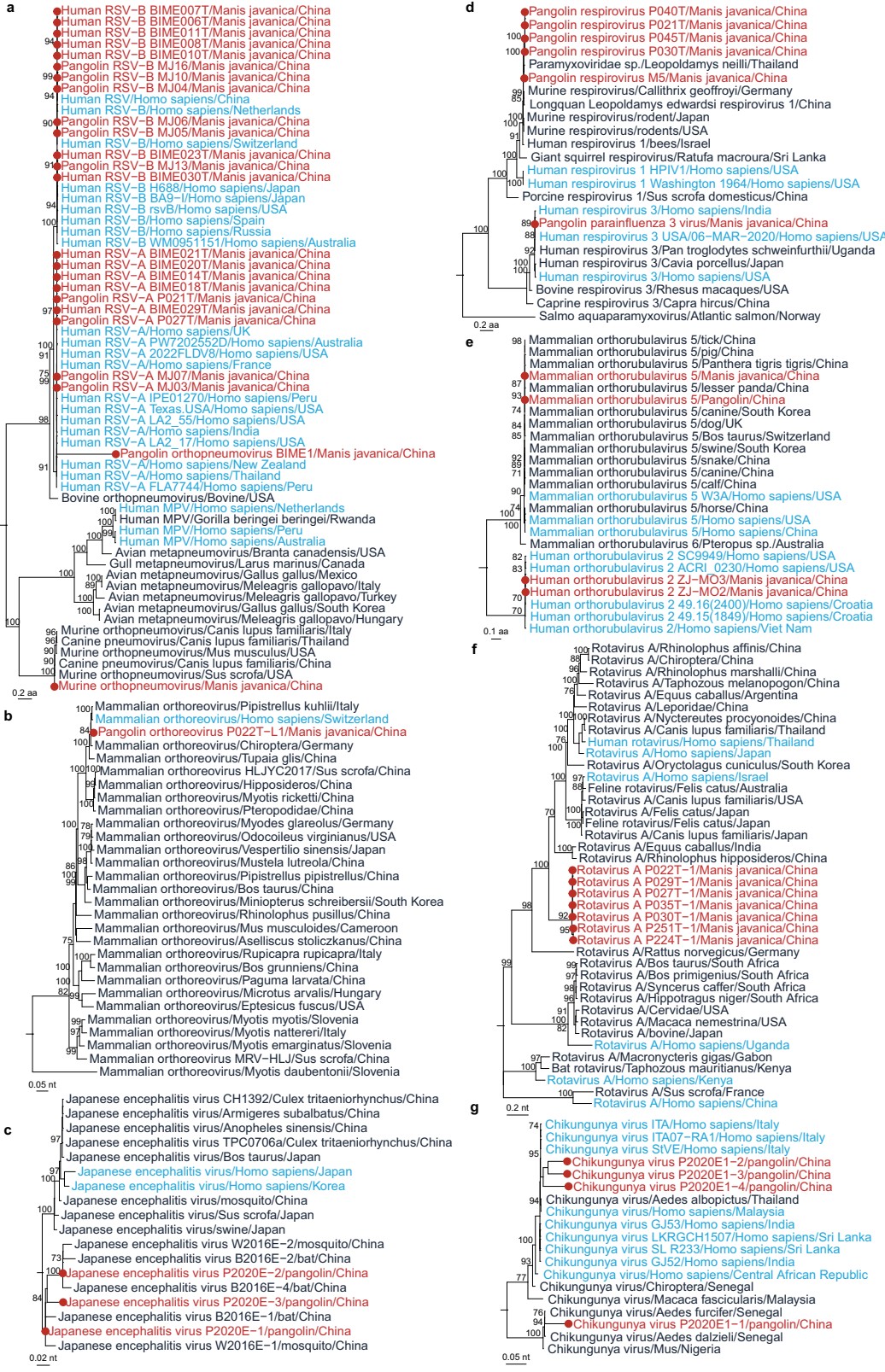

**Fig. 5 | Phylogeny of human-associated viruses in pangolins. a** Phylogeny of viruses in the genus *Orthopneumovirus* based on RNA-dependent RNA polymerase (RdRp) protein (2164 aa). **b** Phylogeny of *mammalian orthoreovirus* based on the RdRp gene (3799 nt). **c** Phylogeny of *Japanese encephalitis virus* based on the E gene (1500 bp). **d** Phylogeny of viruses in the genus *Respirovirus* based on RdRp protein (2220 aa). **e** Phylogeny of genus *Orthorubulavirus* based on the RdRp protein (2255 aa). **f** Phylogeny of *Rotavirus A* based on the VP1 (RdRp) gene (3258 nt). **g** Phylogeny of *Chikungunya virus* based on the CHIKVgp2 gene (1317 nt). Source data are provided as a Source Data file.

name of each pangolin species according to the NCBI Taxonomy Database (https://www.ncbi.nlm.nih.gov/Taxonomy/Browser/www tax.cgi?id=9971), regardless of different common names reported in original publications. Supplementary Table 7 provides a checklist for the Latin and common names of each pangolin. A publication was retrieved if any term was included in its title, abstract, or keywords. Published full articles were included if they were in English or Chinese with reported detailed information on pangolins. Articles were excluded if sufficient information about the distribution or associated microbes of pangolins was unavailable or if they were duplicated. Detailed procedures for literature review and extraction of data are described in the appendix (Supplementary Fig. 15 and Supplementary Text 2). Throughout the paper, we use the term microbes, which are minute, unicellular organisms that are invisible to the naked eye, including viruses, bacteria, and protists. They are also known as microorganisms or microscopic organisms, as they can only be seen under a microscope. We also extracted data on geographical distribution and microbes carried by pangolins from two related websites, including the Global Biodiversity Information Facility (https://www.gbif.org/) and the IUCN Red List of Threatened Species (https://www.iucnredlist.org/en).

The sequences with related information of pangolin-associated microbes were obtained from GenBank (https://www.ncbi.nlm.nih.gov/genbank/), Global Initiative on Sharing All Influenza Data (GISAID, https://www.epicov.org/), and National Genomics Data Center (NGDC, https://ngdc.cncb.ac.cn/) by searching the Latinized (Linnaean) binomial names and alternative names of each pangolin in the three databases. All the sequences of pangolin-associated microbes available in these databases by February 16, 2023, were downloaded, and related information of each sequence was simultaneously extracted, including pangolin species, location, microbe taxonomy, and submission and release dates of the sequence. The duplicated sequences from different databases were excluded. Because information on the sex of pangolins was unavailable in most data sources, we did not carry out any analysis of sex differences in this study.

### Mapping of pangolins and associated microbes
Thematic maps showing the geographical distribution of pangolins and associated microbes were produced with ArcGIS (version 10.6; ESRI, Redlands, CA, USA). According to recorded origins, pangolins were divided into three categories, involving wild, trafficked, and captive pangolins. The captive pangolins were those raised in zoos, rescue centers, or breeding farms. The recorded locations with specific latitude and longitude coordinates were used for mapping. If the exact locations were not available, the centroids of the administrative region were used.

### Meta-analysis of pangolin-infected microbes
In meta-analyses, we included the studies reporting positive numbers or rates of microbes using different detection methods, including PCR, next-generation sequencing and microscopy, and excluded studies that did not report the number of tested and positive samples or percentages that allowed these raw numbers to be calculated. For calculating the positive rate of each microbe family, we used two approaches. If each microbe was explicitly indicated among individual samples in an original publication, we calculated the positive rate of each family by dividing the total number positive for the microbe members in the family by the sample numbers detected. If the publication only provided the overall positive number or rate of each microbe, we could not know whether a single sample positive for more than one microbe of the same family, and thus we used the highest positive rate of the microbe to represent the prevalence of the whole family. Consequently, each study was only included in the meta-analysis once. In case more than one pangolin species was tested for a microbe in the same study, the testing results in different pangolin

species were respectively included in the meta-analysis. We estimated the combined positive rate and 95% confidence interval (CI) of each pangolin-associated microbe. In case only one study was included for a microbe, the positive rate was calculated by the number of positive pangolins divided by the total number of samples tested without a 95% CI. $I^2$ statistic was first estimated to measure the heterogeneity of the data. The fixed effect model was applied if $I^2$ was lower than 50%. Otherwise, the random effects model was used. Meta-regression analyses were performed to explore heterogeneity in relation to the potential factors, including pangolin species, sampling countries, pangolin types (wild, trafficked, and captive), and testing methods (PCR, next-generation sequencing, and microscopy). The meta-analyses were conducted using meta package (v6.5-0) of R (version 4.2.1).

### Characterization of pangolin-associated microbes
We characterized the diversity of microbes among different species of pangolins, as well as in three different categories (wild, trafficked, and captive pangolins). We categorized the microbes into four types according to the associated hosts: (1) Type 1 was the microbes that had been reported in humans; (2) Type 2 was the microbes that had been initially discovered in pangolins and had never been detected in other animals up to date; (3) Type 3 was the microbes that had been reported to infect other animals as well; (4) Type 4 was the unclassified microbe species that had been deposited in GenBank with unknown hosts except for pangolins.

### Phylogenetic analysis of pangolin-associated viruses
Phylogenetic analyses were performed based on the coding sequences (CDS) of the conserved viral proteins, including the RdRp domain for RNA viruses, capsid protein for DNA viruses, and specific genes for viruses that lack the above regions. All sequences were first aligned with related viral sequences within the family using the program MAFFT v7.505 employing the E-INS-i algorithm[20]. All ambiguously aligned regions were then trimmed using the trimAl program (v1.4.rev15)[21]. The trimmed sequence lengths utilized for phylogenetic analysis are listed in Supplementary Table 8. Maximum likelihood (ML) trees were constructed using the IQ-TREE version 2.2.0.3 program[22], employing the best-fit models with 1000 bootstrap replicates. The ggtree (v3.4.4), phangorn (v2.9.0), and ggplot2 (v3.4.2) packages in R software were used to visualize the trees and determine the midpoint as the root of the phylogenetic tree.

### Reporting summary
Further information on research design is available in the Nature Portfolio Reporting Summary linked to this article.

## Data availability
Data were collected from multiple sources, including literature review, related websites, and GenBank, GISAID, and NGDC. The sequences of pangolin-associated microbes used in this study are available in the GenBank (https://www.ncbi.nlm.nih.gov/nuccore/), GISAID (https://www.gisaid.org/), and NGDC (https://ngdc.cncb.ac.cn/) database under accession numbers shown in Supplementary Table 3. Source data are provided with this paper.

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

## Acknowledgements

This study was supported by the Natural Science Foundation of China (82103897, L.Z.), the Natural Science Foundation of Shandong Province, China (ZR2020QH299, L.Z.), Cheeloo Young Scholar Program of Shandong University (L.Z.), and State Key Research Development Program of China (2021YFC2302001, X.-M.C.), and the CAMS Innovation Fund for Medical Sciences (2019-I2M-5-049, W.-C.C.).

## Author contributions

W.-C.C., L.Z., and X.-M.C. conceptualized the initial hypothesis and conceived and designed the study. R.-Z.Y., X.-Y.W., B.-Y.W., and Y.-Y.L. collected the data and conducted the data analysis. K.S., Y.-F.W., J.L., J.-F.J., and B.-H.W. did the statistical analysis. R.-Z.Y., Y.-Y.L., S.-S.W., Q.X., and Z.-H.L. performed phylogenetic analysis and interpretation. Y.-D.D., J.-Y.L., J.-J.Z., L.-F.D., W.S., and N.J. prepared the figures and tables. R.-Z.Y., X.-Y.W., Y.-Y.L., and X.-M.C. wrote the first draft of the manuscript, and W.-C.C. and L.Z. revised the manuscript. All authors contributed substantially to data acquisition and interpretation and revision and editing of the manuscript.

## Competing interests

The authors declare no competing interests.
