## [Peer Review File · Nature Communications]

REVIEWER COMMENTS

Reviewer #1 (Remarks to the Author):

The manuscript summarized the data related to pangolin distribution and microbes from multiple sources including databases, publications and media reports and focused on more virus discovery in pangolins. The results showed that pangolins were infected by multiple viruses which were known to infect humans and other animals. The study suggests that pangolins may play an important role in transmission of viruses. Although the study provides more information, it lacks the detailed information of analyzed sequences and deeper investigation regarding the interspecies transmission risk of pangolin-associated viruses.

Major comment:

In “Phylogenetic analysis of pangolin-associated viruses”, authors please provide detailed information of the sequence length and indicate the minimum length of gene used for phylogenetic analysis and how to identify the authenticity of the sequences.

Minor comments:

1. Lacking page and line numbers throughout the text.
2. Results: first paragraph, please clarify “the number 2335 records” are based on what? Publications? database? Trafficked events? Do the microbes data are among the 2335 records?
3. “112 sequences from 19 families” should be “112 sequences being associated with 19 families”?
4. In “Among the 122 pangolin-associated microbes, 39 (including 13 viruses, 22 bacteria, and four protists) could cause human infections”, “could cause human infections” may be expressed as “have been reported in human”?
5. In “Our phylogenetic analyses revealed that both HRSV-A and HRSV-B in pangolins were same with those currently circulating in humans”, “same” means how much nt/aa identity in genome/gene/protein?
6. In the result description, authors should cite the original reports if there are and clarify what are the newly discoveries by the authors from the data.
7. In “If ICTV lacks clear criteria in the genera, we used a threshold of amino acid identity of 90% for the RNA-dependent RNA polymerase domains (RdRp) (RNA viruses) or capsid proteins (DNA viruses)” is not true. Different families have their own classification criteria, the 90% percent may apply to coronavirus not others, please check it.

Reviewer #2 (Remarks to the Author):

The manuscript presents a meta-analysis of the prevalence of various microbes in Pangolins. It provides highly valuable information and is very well written, with mostly sound methodology. However, there are several issues that need to be addressed, which I will detail below:

1. It appears that the same animals were used multiple times in the meta-analysis to calculate the overall prevalence of certain microbe families. This situation likely arose because several members of the microbe family were detected in the same study. This introduces bias, as the same data are used repeatedly in the analysis. To address this, I suggest recalculating the prevalence of each family using one of the following methods: a) Calculate prevalence based on the presence of at least one tested family member in each study, counting each study only once; or b) Choose one pathogen that is most frequently searched for in the other studies for the meta-analysis calculation. This will help avoid over-weighting studies that tested several members of the same family and will provide more accurate summarized measures with appropriate confidence intervals (CI).

2. I am concerned that there are no studies in the meta-analysis reporting zero prevalence of a certain pathogen tested. It is essential to verify the data and include figures from studies where zero prevalence was documented, if available.

3. The authors should address the evident heterogeneity observed in most of the analyses. To achieve this, they can perform meta-regression analyses, considering factors such as Pangolin species, pangolin types (wild, trafficked, and captive), and location. Explaining this heterogeneity will significantly contribute to the conclusions of the manuscript.

Overall, addressing these comments will enhance the quality and robustness of the meta-analysis and strengthen the conclusions drawn from the findings.

Reviewer #3 (Remarks to the Author):

I have read this manuscript carefully, and the author discusses the microbial community of pangolins using meta-analysis. However, I have some serious concerns about this study.

- (1) Meta-analysis: The sequencing data come from the different sequencing methods. How do the authors avoid the systematic errors during the analysis? The authors should add the details in the method part.
- (2) The taxonomic problem: Some controversies exist in the pangolin's phylogeny. Thus, this is the big problem when the author tries to use the hostname based on some references.
- (3) Positive rate: This study only focuses on the sequencing data. How do we talk about the infection in humans and animals? There are no infection experiments in this study. Thus, most of the parts of the results and discussion are overstated.
- (4) The author should add the details in the classification of the microbes using the different sequences. For example, it isn't easy to make the species identification using part of the 16s data. Also, there are similar problems in virus identification.
- (5) Pangolin-associated microbes: Which part? Skin? Gut? Blood?

Responses to reviewers

Responses to Reviewer #1:

Reviewer #1 (Remarks to the Author):

The manuscript summarized the data related to pangolin distribution and microbes from multiple recourses including databases, publications and media reports and focused on more virus discovery in pangolins. The results showed that pangolins were infected by multiples viruses which were known to infect humans and other animals. The study suggests that pangolins may play important role in transmission of viruses. Although the study provide more information, it lacks the detailed information of analyzed sequences and deeper investigation regarding the interspecies transmission risk of pangolin-associated viruses.

Major comment:

In “Phylogenetic analysis of pangolin-associated viruses”, authors please provide detailed information of the sequence length and indicate the minimum length of gene used for phylogenetic analysis and how to identify the authenticity of the sequences.

Response:

We appreciate the reviewer’s suggestion, and have added a column indicating the trimmed sequence length used for phylogenetic analysis in Supplementary Table 8. Accordingly, we added a statement in Materials and Methods of the revised manuscript as following: “The trimmed sequence lengths utilized for phylogenetic analysis were listed in Supplementary Table 8” (Page 16, Line 375–376). In addition, we have also provided the sequence length used for constructing phylogenetic tree of each virus in the legends of Fig. 5 and Extended Fig. 3.

Minor comments:

1. Lacking page and line numbers throughout the text.

Response:

As suggested by the reviewer, we have added page and line numbers in the revised manuscript.

2. Results: first paragraph, please clarify “the number 2335 records” are based on what? Publications? database? Trafficked events? Do the microbes data are among the 2335 records?

Response:

We appreciate the reviewer's comment, and have clarified the source of the records in the Results as following: "After removing duplication, the analysis of integrated data collected from multiple sources, including literature review, related websites, revealed that a total of 2337 records with locations of different pangolin species were reported in 60 countries around the world, ..." (Page 4, Line 62–65). In fact, we have illuminated how to get the number of 2337 records in the flowchart of Fig. 1, and also described the data collection in Materials and Methods as following: "Data were collected from multiple sources, including literature review, related websites, and GenBank (Fig. 1a). Data from publications were extracted by two independent reviewers (XYW and BYW) through searching electronic databases (PubMed, China National Knowledge Infrastructure, and the WanFang database) for studies published before February 16, 2023" (Page 13, Line 290–294).

As regarding to the data of 142 microbes, 128 microbe records with clear geographic location had been included in the 2337 records. The other 14 microbe records were not included in the distribution analysis of pangolins, due to unavailable information on their location. We have clarified this issue in the Results as following: "Of the 142 microbial records extracted from the publication, 128 with clear geographic location had been included in the 2337 records for distribution analysis, while the other 14 were not included due to unavailable information on their geographic location" (Page 4, Line 74–77).

3. "112 sequences from 19 families" should be "112 sequences being associated with 19 families"?

Response:

As suggested by the reviewer, we have revised the sentence as following: "112 sequences being associated with 19 families..." (Page 6, Line 122–123).

4. In "Among the 122 pangolin-associated microbes, 39 (including 13 viruses, 22 bacteria, and four protists) could cause human infections", "could cause human infections" may be expressed as "have been reported in human"?

Response:

We are grateful for the suggestion, and have revised the expression as suggested (Page 8, Line 162).

5. In "Our phylogenetic analyses revealed that both HRSV-A and HRSV-B in pangolins were same with those currently circulating in humans", "same" means how much nt/aa identity in genome/gene/protein?

Response:

We appreciate the reviewer’s comment, and have revised the statement as following: “Our phylogenetic analyses based on RdRp proteins revealed that HRSV-A and HRSV-B in pangolins were in the same clades with those currently circulating in humans, with aa identities of 99.69–99.95% and 99.86–99.95%, respectively” (Page 9, Line 198–201).

6. In the result description, authors should cite the original reports if there are and clarify what are the newly discoveries by the authors from the data.

Response:

We appreciate the reviewer’s valuable comments, and have add the citations regarding original reports wherever necessary. We have performed a comprehensive analysis based on the integrated data and provided a summary description of the new findings.

7. In “If ICTV lacks clear criteria in the genera, we used a threshold of amino acid identity of 90% for the RNA-dependent RNA polymerase domains (RdRp) (RNA viruses) or capsid proteins (DNA viruses)” is not true. Different families have their own classification criteria, the 90% percent may apply to coronavirus not others, please check it.

Response:

We appreciate the reviewer’s comment, and agree with the point of view on species demarcation criteria of viruses. To avoid this controversy, we retain all the species names reported in original publications in our meta-analysis, and have removed the reclassification for some viruses from Materials and Methods. Accordingly, we have revised the Fig. 1, Fig. 4, Supplementary Table 1–5, and forest plots in Supplementary Fig. 6.

Responses to Reviewer #2:

Reviewer #2 (Remarks to the Author):

The manuscript presents a meta-analysis of the prevalence of various microbes in Pangolins. It provides highly valuable information and is very well written, with mostly sound methodology. However, there are several issues that need to be addressed, which I will detail below:

1. It appears that the same animals were used multiple times in the meta-analysis to calculate the overall prevalence of certain microbe families. This situation likely arose because several members of the microbe family were detected in the same study. This introduces bias, as the same data are used repeatedly in the analysis. To address this, I suggest recalculating the prevalence of each family using one of the following methods: a) Calculate prevalence based on the presence of at least one tested family member in each study, counting each study only once; or b) Choose one pathogen that is most frequently searched for in the other studies for the meta-analysis calculation. This will help avoid over-weighting studies that tested several members of the same family and will provide more accurate summarized measures with appropriate confidence intervals (CI).

Response:

We appreciate the reviewer’s valuable comment, and have re-calculated the prevalence of microbe families according to the reviewer’s the first suggestion, and added the description of calculation method as in Materials and Methods as following: “For calculating the positive rate of each microbe family, we used two approaches. If each microbe was explicitly indicated among individual samples in an original publication, we calculated the positive rate of each family through dividing the total number positive for the microbe members in the family by the sample numbers detected. If the publication only provided the overall positive number or rate of each microbe, we could not know whether a single sample positive for more than one microbe of the same family, and thus we used the highest positive rate of the microbe to represent the prevalence of the whole family. Consequently, each study was only included in the meta-analysis once” (Page 15, Line 339–347). Accordingly, we have updated the results in Fig. 3, Fig. 4, and the forest plots in the Supplementary Information.

2. I am concerned that there are no studies in the meta-analysis reporting zero prevalence of a certain pathogen tested. It is essential to verify the data and include figures from studies where zero prevalence was documented, if available.

Response:

We are grateful for the reviewer’s concern, and have gone through all the included publications, and confirmed that there were six records in five studies reported zero positive rates of *pangolin coronavirus* in *Manis pentadactyla* (Nga et al., Front Public Health, 2022; 10: 826116; Li et al., Front Microbio, 2021; 12: 65743; Peng et al., Zool Res, 2021; 42: 834-4; Xiao et al., Nature, 2020; 583: 286-9;) and *Phataginus tricuspis* (Peng et al., Zool Res, 2021; 42: 834-4), and zero positive rate of a *pangolin pestivirus* (Shi et al., Front Microbiol, 2022; 13: 9887). As suggested by reviewer, we have included the data of these study into the meta-analysis of *pangolin coronavirus* and *pangolin pestivirus* in Supplementary Fig. 5 and Supplementary Fig. 6. The meta-analysis result of *pangolin coronavirus* and

pangolin pestivirus has been presented in Fig. 4. Accordingly, we have revised the number of records in Results as following: “After removing duplication, the analysis of integrated data collected from multiple sources, including literature review, related websites, revealed that a total of 2337 records with locations of different pangolin species were reported in 60 countries around the world, including 734 Chinese pangolin (*Manis pentadactyla*) records, 617 Indian pangolin (*Manis crassicaudata*) records, 479 Malayan pangolin (*Manis javanica*) records,...Of the 142 microbial records extracted from the publication, 128 with clear geographic location had been included in the 2337 records for distribution analysis, while the other 14 were not included due to unavailable information on their geographic location. Among 142 microbial records, 112 were associated with viruses, 11 with bacteria, 13 with protists, and six were records with zero positive rates.” (Page 4, Line 62–79)

3. The authors should address the evident heterogeneity observed in most of the analyses. To achieve this, they can perform meta-regression analyses, considering factors such as Pangolin species, pangolin types (wild, trafficked, and captive), and location. Explaining this heterogeneity will significantly contribute to the conclusions of the manuscript.

Overall, addressing these comments will enhance the quality and robustness of the meta-analysis and strengthen the conclusions drawn from the findings.

Response:

We are grateful for the valuable suggestion, and have performed the meta-regression analyses and added description of the statistic in Materials and Methods as following: “Meta-regression analyses were performed to explore heterogeneity in relation to the potential factors, including pangolin species, sampling countries, pangolin types (wild, trafficked, and captive), and testing methods (PCR, next-generation sequencing, and microscopy)” (Page 15, Line 353–356). According to the meta-regression results, we have added the statement in the Results as following: “To explore the reason for heterogeneity in infection rate of each microbe among publications, we performed the meta-regression analysis and did not find any factor associated with the heterogeneity (Supplementary Table 5–6)” (Page 9, Line 187–190).

Responses to Reviewer #3:

Reviewer #3 (Remarks to the Author):

I have read this manuscript carefully, and the author discusses the microbial community of pangolins using meta-analysis. However, I have some serious concerns

about this study.

(1) Meta-analysis: The sequencing data come from the different sequencing methods. How do the authors avoid the systematic errors during the analysis? The authors should add the details in the method part.

Response:

We appreciate the reviewer’s comments, and have provided a detailed description of different detection methods in Material and Methods as following: “In meta-analyses, we included the studies reporting positive numbers or rates of microbes using different detection methods including PCR, next-generation sequencing and microscopy, and excluded studies that did not report the number of tested and positive samples or percentages that allowed these raw numbers to be calculated.” (Page 15, Line 335–339)

As suggested by reviewer 2, we have also performed the meta-regression analyses to explore the influence of detection methods on heterogeneity among studies, and added description of the new statistic in Materials and Methods as following: “Meta-regression analyses were performed to explore heterogeneity in relation to the potential factors, including pangolin species, sampling countries, pangolin types (wild, trafficked, and captive), and testing methods (PCR, next-generation sequencing, and microscopy)” (Page 15, Line 353–357). According to the meta-regression results, we have added the statement in the Results as following: “To explore the reason for heterogeneity in infection rate of each microbe among publications, we performed the meta-regression analysis and did not find any factor associated with the heterogeneity (Supplementary Table 5–6)” (Page 9, Line 187–190).

Furthermore, we have discussed the limitation regarding to the possible bias owing to the different detection methods as following: “Secondly, the pangolin species without reported pathogens currently might still have microbes which have not been detected before, and variations in detection technology could have introduced bias into the positive rate of pathogens” (Page 12, Line 268–270).

(2) The taxonomic problem: Some controversies exist in the pangolin's phylogeny. Thus, this is the big problem when the author tries to use the hostname based on some references.

Response:

We appreciate the reviewer’s concern, and have clarified the taxonomic approach in Materials and Methods as following: “To avoid controverse taxonomy of pangolins, we utilized the specific Latin name of each pangolin species according to the NCBI Taxonomy Database (<https://www.ncbi.nlm.nih.gov/Taxonomy/Browser/wwwtax.cgi?id=9971>), regardless of different common names reported in original publications.

Supplementary Table 7 provides a check list for Latin and common names of each pangolin” (Page 13, Line 296–301).

Accordingly, we have always provided the Latin name for each pangolin species through the text, regardless of what their common names are to avoid confusion.

(3) Positive rate: This study only focuses on the sequencing data. How do we talk about the infection in humans and animals? There are no infection experiments in this study. Thus, most of the parts of the results and discussion are overstated.

Response:

We appreciate the reviewer’s comment. This study was an analysis of integrated data of the published literature, rather than an experiment study. Therefore, no direct evidence for the infections in human and animals could be provided through such analysis. Although the commonly accepted human-centric approach considers microbes capable of infecting and/or causing disease in humans as human pathogens regardless of their capacity to infect other animals, we have replaced the expressions such as “cause human infections”, “infectious to humans”, “human viruses” and “to infect humans” with “reported in humans” wherever necessary in the revised manuscript to avoid the overstatements regarding infections in humans.

(4) The author should add the details in the classification of the microbes using the different sequences. For example, it isn’t easy to make the species identification using part of the 16s data. Also, there are similar problems in virus identification .

Response:

We appreciate the reviewer’s valuable comment, and agree with the point of view on classification of microbes. To avoid the confusion about species classification, we have corrected the factitious definition based on 16S rRNA sequences and added a category of microbes in Materials and Methods as following: “4) Type 4 was the unclassified microbe species that had been deposited in GenBank with unknown hosts except for pangolins” (Page 16, Line 365–367).

As to virus taxonomy, we retain all the species names reported in original publications in our meta-analysis, and have removed the reclassification for some viruses from Materials and Methods. Accordingly, we have revised the Fig. 1, Fig. 4, Supplementary Table 1–5, and forest plots in Supplementary Fig. 6.

Furthermore, we have also discussed the limitation as following: “Finally, a lot of early studies have provided limited species information about the microbes detected, preventing us from accurately classifying them” (Page 12, Line 270–272).

(5) Pangolin-associated microbes: Which part? Skin? Gut? Blood?

Response:

We appreciate the reviewer’s inquiry, and have provided the detailed information on sample types from which the pangolin-associated microbes were detected in the Results of the revised manuscript as following: “The microbes were detected from different kinds of pangolin samples, 42 of which were from mixed tissues, and various samples including bloods, sera, lungs, livers, spleens, hearts, muscles, faeces, anal swabs, and throat swabs (Supplementary Table 1)” (Page 6, Line 114–117).

REVIEWERS' COMMENTS

Reviewer #1 (Remarks to the Author):

Authors have answered all my questions. I've no more comments.

Reviewer #2 (Remarks to the Author):

All of my comments have been addressed. However, there is still one thing that needs better clarification: I've noticed that in some of the forest plots created for pathogen families, the same reference is included twice in the calculations (e.g. Pangolin Coronavirus, Pangolin Pestivirus, Coronaviridae). Kindly review the entire dataset once more and rectify this issue. If there's anything I'm not comprehending correctly, please provide further clarification.

Reviewer #3 (Remarks to the Author):

I appreciate the authors' efforts in this revision. However, the authors didn't address my previous comments. There are some big limitations in this meta-analysis.

REVIEWERS' COMMENTS

Reviewer #1 (Remarks to the Author):

Authors have answered all my questions. I've no more comments.

Reviewer #2 (Remarks to the Author):

All of my comments have been addressed. However, there is still one thing that needs better clarification: I've noticed that in some of the forest plots created for pathogen families, the same reference is included twice in the calculations (e.g. Pangolin Coronavirus, Pangolin Pestivirus, Coronaviridae). Kindly review the entire dataset once more and rectify this issue. If there's anything I'm not comprehending correctly, please provide further clarification.

Response:

We appreciate the reviewer's concern, and have gone through the dataset carefully. In our forest plots, each microbe detected in pangolin species was included once in the meta-analysis to calculate the positive rate. To clarify this issue, we have added a statement in the Methods as following: "In case more than one pangolin species were tested for a microbe in the same study, the testing results in different pangolin species were respectively included in the meta-analysis." (Page 15, line 357–359)

Reviewer #3 (Remarks to the Author):

I appreciate the authors' efforts in this revision. However, the authors didn't address my previous comments. There are some big limitations in this meta-analysis.

Response:

We appreciate the reviewer's concerns. Also according to the editorial guidance, we have discussed further limitation of the meta-analysis in the revised manuscript as following: "Secondly, the pangolin species without reported pathogens currently might still have microbes that have not been detected before, and variations in detection technology could have introduced bias into the positive rate of pathogens. Thirdly, the controversial taxonomy of pangolins due to local names used in the early literatures might impact the geographical distribution objectivity of some pangolin species. Finally, a lot of early studies have provided limited species information about the microbes detected, preventing us from accurately classifying them." (Page 12, line 273–279)